# A Environments

## A.1 DAG environment

The causal structure is sampled according to the following process:

1. An ordering of the variables is picked at random. (At test time, the test goal variable is constrained to be at the end of the order, but the order is otherwise unconstrained.)

2. A noise variance for each of the variables is sampled, from a (clipped) normal distribution with mean 0.5 and standard deviation 0.25.

3. (In the adaptive experimentation version from Section 5.1.1, a subset of half the variables are sampled to be relevant; all subsequent steps only include variables from this subset.)

4. A number of independent variables $i$ is sampled, and the first $i$ variables in the ordering are set to be independent.

5. For the remaining nodes, their input weights are sampled as follows:

   - 1 or 2 nodes from earlier in the order are sampled to be parents of this node. (In training, the test goal variable is constrained to not have parents that are descendents of the test intervention target. At test time, it is selected to *only* have parents that are descendants of the test intervention target.)
   - The input weight from each parent is sampled uniformly in $[-2, 2]$.
   - At test time, all weights on the path from the intervention target to the goal are increased in magnitude to larger than 1, to ensure that intervening on the target (or its ancestors, depending on the test condition) is the optimal way to maximize the goal node. All weights from other inputs to any nodes along the path are decreased to less than 1.

These constraints allow a total of 1,277 possible DAG structures for training.

When propagating values through the graph, each node's value is computed as a linear function of its ancestors outputs, followed by the addition of a value sampled from a normal distribution with mean 0 and the current noise variance of this node, finally followed by a nonlinearity. We used a leaky relu nonlinearity [48] with a leak current of 0.2; we also compare to the case without a nonlinearity in Appendix B.1.

**Observations:** The environment observations contain the following features:

- The values of the variables before the previous intervention.
- The previous intervention taken (as a numerical value of the intervention on the variable it was applied to, e.g. $[0, 0, -4, 0, 0]$).
- The outcome values after the previous intervention.
- The initial values for the next experiment.
- The goal variable to maximize, as a one-hot value (or zeros before the goal is presented).
- (Only in the adaptive experimentation conditions, a several-hot vector indicating which variables are relevant.)

The environment offers a discrete action space of size $2 \times n$, corresponding to the two possible interventions ($-4$ or $4$) that the agent can perform on any of the $n$ variables.

**Expert policy:** In the exploration phase, the expert policy randomly chooses a variable to intervene on first, then intervenes on the variables in order after that, looping back to 0 after it reaches the end; e.g., the expert intervention order might be [2, 3, 4, 0, 1]. Whether the expert interventions are positive or negative is chosen randomly on each step. In the exploitation phase, the expert takes the optimal action to increase the value of the goal variable.

**Heuristic baseline policies:** We evaluate the baselines only during the goal-seeking/exploitation phase. The value baseline tracks the outcomes of interventions during the exploration phase, then at exploitation time takes the action which gave the highest value on the target node. The change baseline is similar, but sign sensitive; it chooses the intervention that changed the value of the goal

node the most, and then flips the sign of the intervention if the original change in the outcome was negative.

**Correlation baseline policies:** The correlation baselines constructs a policy based on the correlational statistics of the observations the agent makes during training; by fitting linear regression models. WLOG, assume the target node is E, and there are $n = 5$ nodes. The total correlation baseline fits $n - 1$ separate univariate regressions, one from A to E, one from B to E, etc. The partial correlation baseline fits a single multivariate regression from $A, B, C, D \rightarrow E$; that is, it controls for the effect of the other variables. In either case, the variable with the largest regression slope is chosen to intervene on, with a sign corresponding to the sign of the slope.

## A.2 Odd-one-out environments

We use the open-sourced version of the odd-one-out environments [41], which is available at: `https://github.com/deepmind/tell_me_why_explanations_rl`.

**Feature combination split:** We simply split combinations of features randomly into a train and test set by their hash value; we hold out 20% of the hash values for testing. This is a simple split, so it is not particularly surprising that the agent generalization performance is comparable to train performance.

**Hard dimension split:** The odd-one-out environments afford several possible initialization of the objects before the experiments, that affect how hard the experiments are to perform (Fig. 6). We train the agent on the three easier initializations (with 0-2 dimensions having the hard setting), and test on the case where all three do.

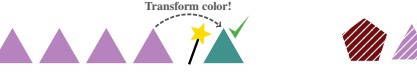

(a) Easiest initialization.          (b) Hardest initialization.

Figure 6: Cartoon examples of possible object sets from the (a) easiest train, and the (b) hard test level in the odd-one-out dimension split tasks. In the simplest case, the agent can choose the object it has just transformed. However, in the harder levels, the agent has to choose a *different* object, which is left unique after its transformation, in order to test a dimension. Choosing the object it transformed will always be incorrect. Note that these illustrations are just cartoons; the agent still saw pixel observations like those shown in Fig. 3a. (Adapted with permission from Lampinen et al. [41].)

**Expert policy:** We augmented these environments with a hand-designed expert policy which first navigates towards the center of the room until objects come into view, then goes to the nearest object, transforms a feature of it, and then goes to the object which is made unique (not necessarily the transformed one), and chooses it. The expert policy experiments on the three possible feature dimensions (color, shape, and texture) in order. Once it has observed the correct dimension, it continues to choose that (rather than experimenting on the others). On the final trial, it navigates to the object which is unique along the correct dimension. The expert's navigation policy tries to move towards its navigation target as efficiently as possible, with simple one-step heuristics for navigating around obstacles in the way (first try navigating around in a way that brings you closer to your target, if possible; otherwise fall back to successively more roundabout routes).

### A.2.1 Language model adaptation of the odd-one-out intervention tasks

In order to adapt the odd-one-out tasks to the language models, we converted the observations and actions into language. We also reduced the number of objects to three, used the simple contrast structure (all objects initialized the same in the experimenting episodes), and removed the locations and navigation (which are more difficult to describe, and do not contribute substantially to the conceptual issues). We provide a full example episode from the expert policy in Listing 1.

In an evaluation episode for the language model, it was given a four shot prompt of examples of this type (separated by lines of equals signs and "New game:" to delimit a new episode). It would first receive an observation of the first objects, and would be prompted with this followed by "I transform object" and given a choice between A, B and C. The object assigned the highest likelihood

Listing 1: Example expert episode in the LM odd-one-out environment

```
New game:

There is a set of three objects in front of me:
A) white square striped
B) white square striped
C) white square striped

I transform object A into a different color: pink.
Choosing object A was not rewarded.
Explanation: The rewarding dimension must not be color.

There is a set of three objects in front of me:
A) black ellipse striped
B) black ellipse striped
C) black ellipse striped

I transform object B into a different shape: trapezoid.
Choosing object B was rewarded!
Explanation: In this game, I am rewarded for unique shape.

There is a set of three objects in front of me:
A) red pentagon solid
B) red pentagon solid
C) red pentagon solid

I transform object C into a different texture: striped.
Choosing object C was not rewarded.
Explanation: The rewarding dimension must not be texture.

There is a set of three objects in front of me:
A) purple ellipse solid
B) green trapezoid solid
C) green ellipse striped

Reasoning: Let's think step by step. In this game, I am rewarded for
    ↪ unique shape. Object B is the only trapezoid object, because A
    ↪ and C are ellipse, so B has a unique shape. I will be rewarded
    ↪ for choosing object B.
I choose object B
Choosing object B was rewarded!
Explanation: I was rewarded for unique shape in this game.
```

as a continuation would be chosen. Then, the prompt was continued through "into a different" and the model was given a choice between the different possible dimensions (color, shape, and texture). Again, the highest-likelihood continuation was chosen. The environment would then transform the object according to its rules, and the resulting attribute would be presented to the agent. Because we are using the simpler attribute relations, we assumed that the model chose the object it had just transformed. The model was then presented with the reward (or lack thereof) from this experiment.

After each trial, the model was prompted with "Explanation:" and allowed to generate until it produced a newline character. This was inserted into the prompt, followed by the beginning of the next trial. (For the few shot examples, we used explanations that connected the outcome of the experiment to the latent task structure, as illustrated above.)

On the final trial, the language model was presented with three distinct objects, each of which was unique along one of the three dimensions, in a random order. The LM was then prompted with "Reasoning: Let's think step by step." [cf. 35] and again allowed to generate until a newline. (During training, we provided a fixed reasoning format, as illustrated above.) This reasoning trace was appended to the prompt, followed by "I choose object" and a choice between A, B, and C, which was then scored for whether it was correct along the relevant dimension.

**Expert policy:** We found that the expert policy affected generalization — in particular, it was important for the expert to use a fixed experimentation policy (always experiment on first color, then shape, then texture); see Appendix B.8.

**Prompt construction:** We constructed the four-shot prompt for each condition by sampling 10 possible 4-shot prompts, then evaluating each of them on 20 fixed episodes sampled from the train distribution — i.e. if texture dimension was held out, these 20 episodes would involve color or shape. The same 20 episodes were used for evaluating each of the 10 prompts. We selected the prompt that achieved the highest score on this validation set to use for evaluation in the hold-out condition.

# B    Supplemental experimental results

## B.1    The linear causal DAG case

We also removed the nonlinearities and evaluated the case of a fully-linear causal DAG, which makes the learning problem slightly simpler, and makes the correlational strategies stronger baselines. Indeed, if the noise variances on the nodes were fixed, the causal structure could be recovered from observational data alone in this special case [63]. Correspondingly, the baseline approaches match the optimal expert more closely in this setting. Nevertheless, the agent still matches the expert somewhat better than it matches the baselines, even in this case.

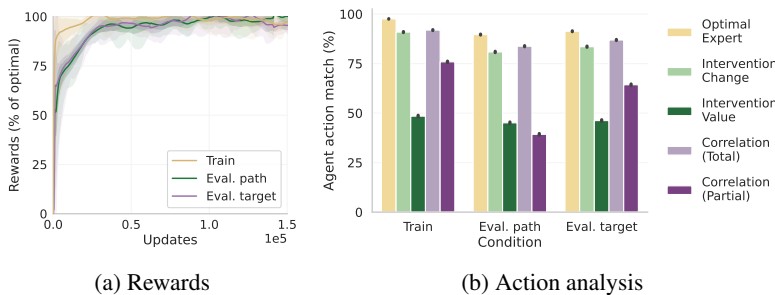

(a) Rewards                                (b) Action analysis

Figure 7: Experimental results in the linear version of the causal DAG environment. (a) Rewards obtained by the agent when evaluated in interactive settings, as a percentage of the optimal rewards. (b) The proportion of the agent's actions that match the optimal behavior, or baselines based on heuristics from the interventions or correlational statistics. While the baselines are more effective in the fully-linear case, the agent still matches the expert strategy significantly more closely than it matches the heuristic baselines. (Error bars are 95%-CIs.)

## B.2    The test-time exploration phase, with active intervention, is critical

The main text emphasizes the importance of agents intervening at test time; however, is this actually necessary? In these ablation experiments, we test this question by removing the active intervention exploration phase from both training and testing, either skipping it entirely or by replacing it with a purely observational phase, where the agent cannot intervene, but merely observes samples from the natural distribution of the causal DAG. Both ablation agents perform much worse than agents that can intervene (Fig. 8), thus demonstrating the importance of intervention at test time to determine the new causal structure.

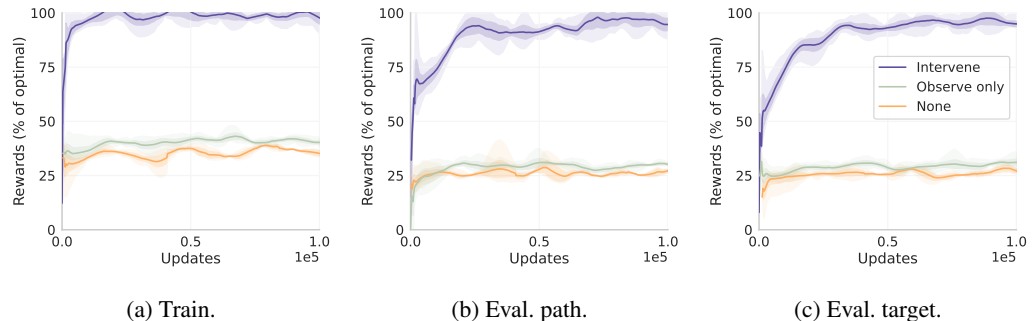

(a) Train.

(b) Eval. path.

(c) Eval. target.

Figure 8: Actively exploring at evaluation time is necessary for effective performance. In these ablation experiments, we remove the active (interventional) exploration phase, either by skipping it entirely (grey-green curves) or by replacing it with a purely observational phase, where the agent cannot intervene, but merely observes samples from the natural distribution of the causal DAG. Compared to agents that can intervene to explore (purple), as in the main experiments, the ablation conditions perform much worse on both (a) graphs sampled from the training distribution and (b-c) graphs sampled from the evaluation distributions. Observing samples from the graph is better than nothing, but still results in much lower performance than intervening.

## B.3  Generalization degrades if agents are trained on few DAGs structures

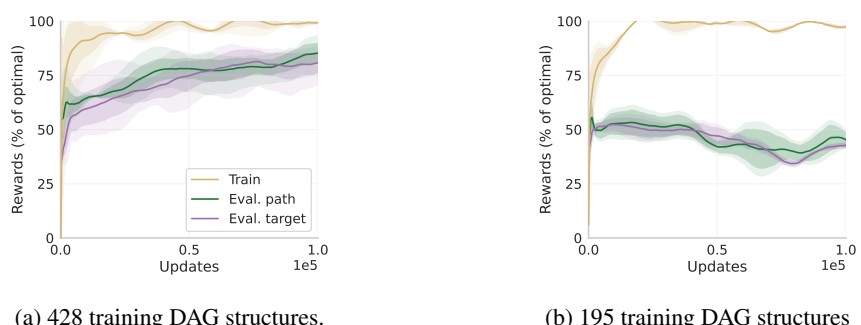

(a) 428 training DAG structures.

(b) 195 training DAG structures

Figure 9: Exploring agent generalization under more restrictive training DAG structure holdouts. In these experiments we impose additional restrictions on the training DAGs, while maintaining the same evaluation set as the original experiments. While the original experiments used 1277 training DAG structures (though of course each structure can be instantiated with infinitely many weight patterns), we impose additional dependency constraints that reduce these to (a) 428 or (b) 195 training DAG structures. In the former case, we still seem some generalization, though it is lower than the original experiments. In the second case generalization performance is quite poor, and we observe evidence of overfitting as training progresses.

In this section, we evaluate how agents would perform if they were trained with fewer DAG structures. In the original experiments we held out causal structures that would make node D an ancestor of E. In combination with the other constraints we impose on the graph structures this results in a total of 1277 distinct DAG structures that can occur in training (although of course each structure can in turn be instantiated with many different combinations of edge weights). Here, we reduce this number of possible structures further, by imposing additional constraints:

- That neither A nor D can be an ancestor of C or E. This results in 428 possible training DAG structures.
- That none of A, D, or E can be an ancestor of B, C, or E. This results in 195 possible training DAG structures.

The results of these experiments are shown in Fig. 9. As we impose more restrictions on the training distribution, we observe correspondingly weaker generalization. In the first case (428 training

structures), we still observe some generalization to novel structures, but in the second case (195 training structures), we observe substantial overfitting.

## B.4 Generalizing to functionally different causal DAGs

In this experiment, we explored whether agents strategies would be robust to a shift between the kind of causal DAGS experienced in training and evaluation. We considered two shifts:

1. Training on causal DAGs with nonlinearities after the sum on the node (as in the main experiments), then evaluating on DAGs that additionally introduce (leaky ReLU) nonlinearities on the edges before the sum.

2. Training on linear DAGs (as above in Appendix B.1), and then evaluating on nonlinear DAGs (as in the main experiments).

Fig. 10 shows the results. Agent strategies reveal some robustness to these shifts. The agents show very strong generalization in the case of introducing nonlinear edges (though this may be because the nonlinearities on the node mask the effect of edge nonlinearities). In the case of generalizing from linear to nonlinear graphs, there is a larger shift, and so generalization is less perfect, but agents still achieve over 75% of optimal performance.

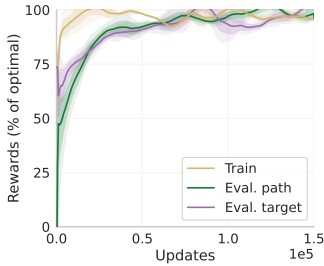
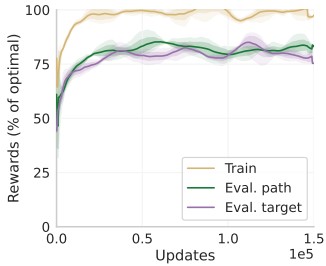

(a) Generalizing to nonlinear edges.      (b) Generalizing linear to nonlinear.

Figure 10: Agents can generalize their causal strategies to different functions types within the causal DAG. In addition to the training/evaluation split used in the main experiments, we introduce an additional shift in evaluation by changing the functions within the graph. (a) Introducing nonlinearities on the edges in evaluation, in addition to the nonlinearities on the nodes. This has relatively little effect on test performance (perhaps because the node nonlinearities effectively mask most the effect of the edge ones). (b) Training on linear causal DAGs, and evaluating on nonlinear ones. This evaluation induces a more dramatic shift, and results in a more noticeable generalization gap. However, agents still demonstrate some robustness, achieving over 75% of optimal performance.

## B.5 Adaptive experimentation — further results

In this section, we show further analysis of the agent performance from the adaptive experimentation strategies (Section 5.1.1) — specifically, we show that the agent achieves high evaluation rewards, and continues to match the optimal policy much more closely than the baselines.

It might seem surprising that in this setting the correlation baselines perform very poorly. However this is due to the statistics of the particular situation. Because the number of fully independent variables — the 5 irrelevant ones — is equal to the number of interventions performed (and interventions on any other variable do not, of course, affect the irrelevant variables, and often do not affect the target either), there is a relatively high probability that at least one of the independent variables will have a non-trivial correlation with the target. Furthermore, because the variance of the irrelevant nodes is lower on average (as they have no inputs), their correlation is effectively *amplified* in the regression slope, so they end up generally having larger slopes than the actually relevant variables.

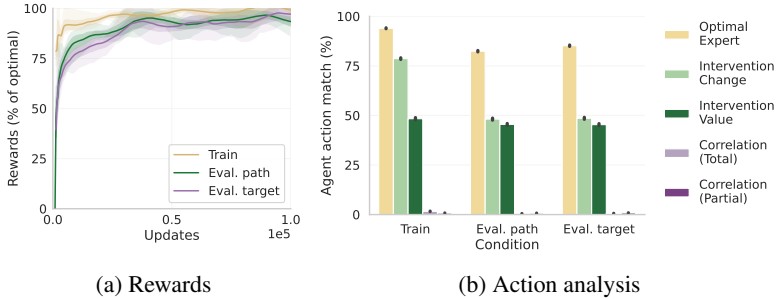

(a) Rewards                    (b) Action analysis

Figure 11: In the causal DAG environment with adaptive experimentation strategies, (a) the agent continues to achieve high rewards in evaluation, and (b) the agent continues to more closely align with the optimal policy than with heuristic baselines.

## B.6   Explanations are key for learning & generalization in the odd-one-out environments

In Fig. 12 we show that agents learn much more effectively with explanations in the odd-one-out environments. Analogously, in the language model experiments explanations were key for generalizing these task structures, as we explore in more detail in the next section.

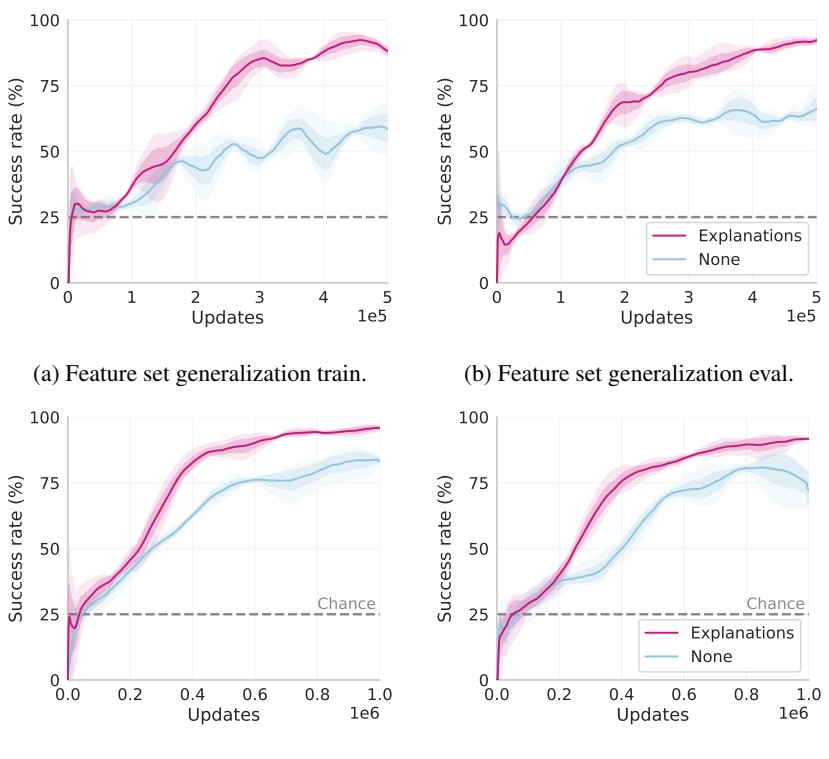

(a) Feature set generalization train.        (b) Feature set generalization eval.

(c) Hard dimension generalization train.      (d) Hard dimension generalization eval.

Figure 12: Explanations are key for the basic agent to learn effectively in the odd-one-out environments. Across the two generalization splits: feature splits (top) and hard dimension generalization (bottom), and across train (left) and eval (right) in each case, the agent performs significantly better when trained with explanations.

## B.7   Exploring how prompts affect LM generalization

Here, we evaluate language model performance in more detail, across a range of prompt conditions. In addition to the versions with explanations and chain-of-thought reasoning and without either,

which we included in the main text, we consider prompts with explanations only, reasoning traces only, or a detailed explanatory instruction included at the beginning of each episode:

```
In this game, I must choose an object that is unique along the "
    ↪ correct" dimension. Before choosing an option, I can
    ↪ sometimes transform one of the objects to make it unique
    ↪ along some dimension, not necessarily the correct one:
```

In Fig. 13 we show that the LMs can generalize in the odd one out task given either explanations or chain-of-thought reasoning alone; however, performance may be more reliable with both.

In Fig. 14 we further break these results down by dimension that is held out. Intriguingly, there are striking dfferences in generalization performance on the different dimensions; while prompts with explanations and/or reasoning tend to yield strong generalization across all three, color and shape appear more difficult than texture. These feature biases may be related to those observed in prior work [9].

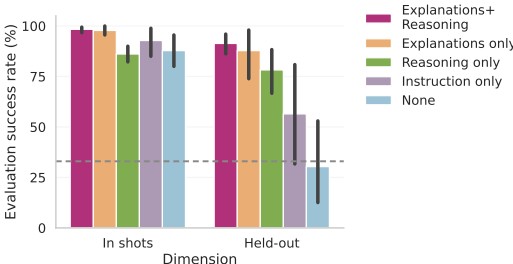

Figure 13: Aggregate LM performance across different prompt conditions. Either explanations or chain-of-thought reasoning are sufficient to substantially improve LM average performance on the odd-one-out tasks; however, having both results in more reliable generalization. Instructions may also result in some generalization.

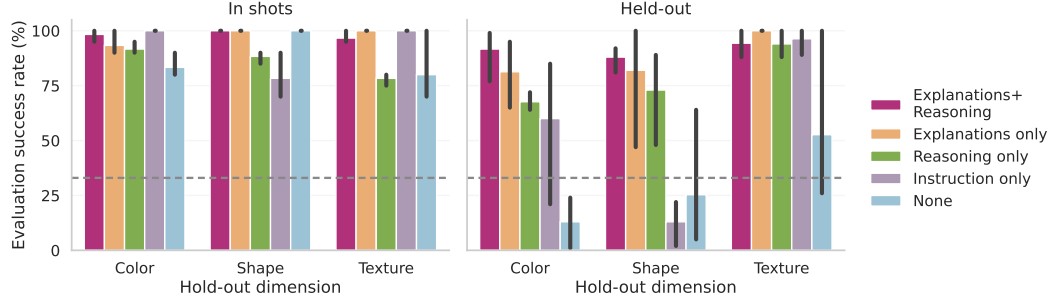

Figure 14: LM performance across different prompt conditions, broken down by dimension. There are noticeable differences in generalization across the different dimensions, but the overall pattern of advantages for explanations + reasoning generally holds.

## B.8   Language models generalize less well from a more complex experimentation policy

In the main text, the expert policy for the LM experiments followed a fixed order (color, shape, texture). Here, we explore LM performance if the expert policy in the examples is more complex: it follows the fixed order until it discovers the correct dimension, but then switches to exploiting this knowledge, rather than continuing with its fixed experiments.

We show the results in Fig. 15. Performance is noticeably worse, particularly if texture is held out. Texture is the final dimension in the fixed order, so if texture is held out, none of the example shots will contain more than one failed experiment, or any examples of trying texture at all. In this case, the language model performs somewhat less well at new samples even from the dimensions included

in the shots (color and shape), and performs drastically worse in held-out evaluation. These results suggest that it is beneficial for the language model to see some shots with multiple experiment failures — even to perform well on new samples from the color & shape train set, where multiple failures will not occur if it follows the same exploration order — and certainly to generalize to the case where texture is held out. It is possible, of course, that more examples in the prompt could overcome this challenge.

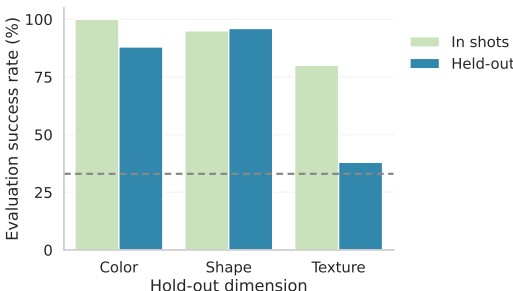

Figure 15: Language models generalize less well from an more complex expert policy — if the expert policy switches to exploiting after it discovers the correct dimension, the language model does not learn or generalize as well in the texture case. (Compare to the results where the expert follows a fixed policy, shown in Fig. 5.)

### B.9 LMs can generalize the odd-one-out tasks even if an extra object is added to the test trial

Evaluating the LM with a surprise extra test object. That is, we kept the prompts the same as the above experiments (three objects in the experimentation and test trials). However, in addition to evaluating on a held-out dimension, we introduced a fourth object in the final test trial of the evaluation episode. More specifically, we kept three objects in the experimentation trials of the test episode, and only surprised the model with a fourth object (which was not unique along any dimension) in the final choice of objects. This experiment tests whether the LM can generalize based on uniqueness when both the relevant dimension and the object distribution changes from training to test.

The results are shown in Fig. 16. While the added object does reduce generalization performance somewhat relative to the original three-object version, performance is still far above chance (25%). Thus, the LMs appear to generalize relatively robustly when prompted with explanations, even if the test trial differs from the training in several ways.

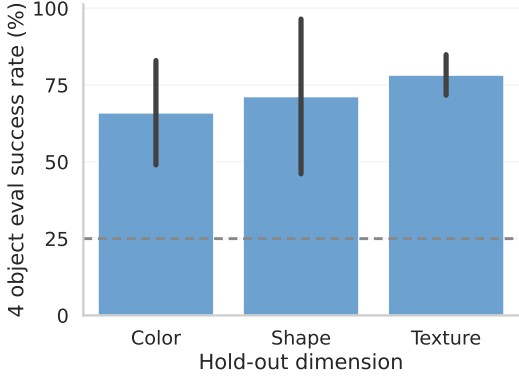

Figure 16: LM evaluation performance on held-out dimensions, as above, but with an extra surprise object in the final test trial. The LM performs far above chance even in this more challenging test condition.

## C   Supplemental methods

### C.1   Agent hyperparameters & training

In Table 1 we list the architectural and hyperparameters used for the main experiments. All agents were implemented using JAX [7] and Haiku [26].

Table 1: Hyperparameters used for the agents.

| | Causal DAG | Odd-one-out |
|---|---|---|
| Memory feature dimension | | 512 |
| Memory layers | | 4 |
| Memory num. heads | | 8 |
| TrXL extra length | 16 | 128 |
| Visual encoder | | CNN |
| Vis. enc. channels | | (16, 32, 32) |
| Vis. enc. filt. size | | (9, 3, 3) |
| Vis. enc. filt. stride | | (9, 1, 1) |
| Policy | | Single linear layer. |
| Explanation decoder | | 1-layer LSTM, embedding size 128 |
| BC loss weight | | 1. |
| Explanation loss weight | | 0.1 |
| Batch size | | 32 |
| Training trajectory length | 20 | 50 |
| Optimizer | | Adam [33] |
| LR | $2 \cdot 10^{-4}$ | $1 \cdot 10^{-4}$ |
| Max gradient norm | 10 | 1 |

Hyperparameters were not extensively tuned; they were mostly set to the same values from prior work [41], with the following exceptions:

- We increased the weight on the explanations loss in the intervention odd-one-out environments (to 0.1), to better match the magnitude of the BC loss; learning was slightly impaired with a smaller value of the explanation loss.

- Learning in the intervention odd-one-out environments was unstable in our initial experiments — the loss would sometimes become NaN — so we decreased the max value to which the gradient norm was clipped, which seemed to resolve the issue.

- We used shorter trajectory length in the simple causal DAG environment.

- We used a simpler policy architecture (single linear layer).

### C.2   Compute

All agents were trained using a $2 \times 2$ TPU v2 or v3 slice. Language model inference was done on $2 \times 4$ TPU v4 slices. Total usage across all experiments (both agents and LMs), including preliminary runs etc., was approximately 450 TPU-hours.