# OpenReview forum: "Passive learning of active causal strategies in agents and language models"
_NeurIPS.cc/2023/Conference — NeurIPS 2023 poster_

### Official Review · Reviewer_U5ax · 2023-07-03

**Soundness:** 3 good
**Presentation:** 4 excellent
**Contribution:** 3 good
**Rating:** 7
**Confidence:** 3

**Summary:**

This paper provides a thorough analysis of whether agents trained to imitate passive data can indeed learn active strategies for intervening during evaluation. In particular, this paper demonstrates that agents can "learn" causal structure from passive data in that they correctly intervene on the appropriate variables during testing, even if such interventions were not encountered during training. A formal "existence proof" for active causal learning is given. Experiments are demonstrated on a causal DAG environment, a pixel-based shapes environment, and few-shot prompting of a language model.

**Strengths:**

*Originality:* The paper provides an interesting demonstration that refutes some claims from the causality literature. In particular, the paper shows that "it is possible to learn generalizable strategies for causal experimentation and intervention from passive data alone" given that the data was generated from an expert's actions.

*Quality:* the experiments appear technically sound and are well-executed.

*Clarity:* Overall the paper is well-written, with understandable diagrams + captions. Moreover, the flow of the paper is impressive, with the authors smoothly anticipating and responding to potential concerns.

*Significance:* The paper qualifies some claims surrounding agents' abilities to learn active causal strategies from passive data. In particular, this extends our understanding of the limits of agent learning. Moreover, this may amplify the concerns for safety of such agents, as they may appear to learn generalizable reasoning strategies even through passive data.

**Weaknesses:**

While overall the paper is technically sound, the first claim made in the contribution seems a bit too strong when compared against the setting given by the proof. In particular, the authors claim "We formally explain why it is possible for an agent to learn a generalizable strategy for exploiting causal structure from passive data, as long as the agent can intervene at test time". However, their proof relies on an assumption that the system is accurately represented by a finite DAG, which may be unrepresentative of real-world scenarios, which may include cycles due to feedback loops. It would be more accurate to characterize their contribution as an "intuitive explanation of why agents may be able to learn generalizable strategies for exploiting causal structure under certain settings, namely that the system can be represented as a finite DAG."


**Questions:**

Were any ablations done to study how the coverage of the training distribution affects the ability of the agent to learn causal strategies? For example, in the adaptive experimentation setting in the DAG environment the expert policy only explores all the relevant nodes. What happens if the expert policy explores more nodes than the causally relevant nodes? The agent still has enough information to identify the causal structure, but it would be interesting to see if it is able to behave optimally in this case.

More generally, do the authors expect the learning of active causal strategies from passive data to be efficient when presented with more complex environments?

**Limitations:**

The limitations are adequately addressed.

---

> ### Author Rebuttal · Authors · 2023-08-08
>
> We thank the reviewer for their positive assessment of the paper and the results. We agree with the reviewer’s comment about the phrasing of the claims, and we have updated the statements in the abstract and introduction accordingly. We have also added a paragraph to the discussion explicitly focused on the simplicity of the environments and their causal structures (the limitations of finite DAGs, as well as some of the comments from the other reviewers), and noting the possibility that results would be different in more complex settings. We also performed some additional experiments to evaluate generalization to different kinds of causal DAGs (e.g. linear to nonlinear), which show some robustness of the learned strategies, as well as an experiment on generalization of the LLMs to an unexpected extra object in evaluation.
>
> Finally, we performed an experiment where we explored coverage of the training distribution in a slightly different sense, by holding out successively more DAG structures than in the original experiments. In the original experiments, there are 1,277 possible DAG structures that are consistent with the training graph sampling constraints (out of ~29,000 possible DAGs over 5 nodes). We ran two more training runs where we imposed either 3 or 7 more non-dependency constraints on the graph, which reduces the number of possible training DAGs to 428 or 195, respectively. With 428 training DAGs, agents still generalize reasonably well, but with 195 they overfit quite strongly. (Fig. 14 in the global response PDF.) This suggests that some breadth of coverage is required for generalization, but that it is not necessary to thoroughly cover the full distribution.

---

> > ### Comment · Reviewer_U5ax · 2023-08-13
> > **Thank you for the response.**
> >
> > I appreciate the changes and would like to keep my score.

---

### Official Review · Reviewer_KP4B · 2023-07-06

**Soundness:** 4 excellent
**Presentation:** 4 excellent
**Contribution:** 4 excellent
**Rating:** 8
**Confidence:** 5

**Summary:**

The paper studies the possibility of learning to identify new casual structures through purely passive learning, which entails the feasibility of acquiring robust task performance in novel conditions through offline learning. The authors hypothesize that it is possible to learn generalizeable causality exploration strategies from offline data that demonstrate such strategies. An agent can then apply these strategies to explore the casual structures of new environments and effectively achieve goals in them. Experiments on several synthetic environments show that explanations play a crucial role in enabling learning generalizeable exploration strategies. These results provide further insights on the ability of LLMs to quickly acquire interactive skills, even when trained exclusively on offline data.

**Strengths:**

- The paper studies a fundamental question whose answer can have profound impact on the understanding and development of AI agents. It challenges a long-standing assumption about the limitations of learning from observational data. It suggests that observational data are not all the same; certain forms observations can be more useful than others. The new understanding is powerful. It can help explain the recent successes of LLMs, which are mostly trained through observations of the world. More general, it offer valuable insights to better leverage offline data to reduce the cost of collecting interactive data.
- The experiments are well-designed and the analyses are insightful. The results can lead to further theoretical and empirical investigation, especially on the role of the explanations in learning.
- The paper is very clear and well-written. The arguments are nicely presented.

**Weaknesses:**

A major weakness of the paper is that it is mostly empirical.

While the authors provides intuition on why their hypothesis can be true, the paper is still missing a satisfactory theoretical explanation. The environments studied, while having interesting structures, are artificially constructed. The paper may further benefit from having experiments on real-world tasks. The artificial environment may have odd properties that allow generalization to occur.

The main hypothesis rests on the assumption that an agent can learn generalizeable exploration strategies. While the empirical results support this assumption, it is unclear why it should hold in theory, because the strategy demonstrations are also observational data. Why is it easier to learn generalizeable exploration strategies than directly learning generalizeable exploitation strategies? I'd strongly suggest the authors add ablation experiments that dissect the importance of the exploration phase (i.e. results without exploration). This would strengthen the claim that the exploration phase is needed.

The authors emphasize the role of explanations in acquiring robust exploration strategies, but they neither clearly define "explanations" nor show examples in the main paper. This term is quite general and can mean many different things. While some examples are shown in the appendix, I think it should be more explicit in the main paper what type of explanations are used, what causal structures/properties they describe. I would also like to see more discussions or experiments on the effect of the explanation medium on the performance. Do explanations have to be in natural language? Or do they just need to have certain properties?

Related work:

[1] Interactive learning from activity description: provides descriptions of activities as feedback to agents, leading to more efficient and effective learning than RL.

[2] Distilling Step-by-Step! Outperforming Larger Language Models with Less Training Data and Smaller Model Sizes: shows that learning to predict explanations can enhance the pre-training of LLMs.

**Questions:**

See my "weakness" section.

**Limitations:**

The authors compose a very thoughtful and thorough discussion section. I like that the authors anticipate and warn readers to avoid mistakenly interpret the claims made in the paper as passive learning can be as good as active learning.

---

> ### Author Rebuttal · Authors · 2023-08-08
>
> We thank the reviewer for their thoughtful review; we are glad that they found the paper interesting, and we found their suggestions very useful. We have attempted to address them as follows:
> * We have performed a set of ablation experiments that remove the exploration phase in two ways; performance is substantially degraded in both (results are plotted in the response PDF). These results demonstrate the importance of the exploration phase.
>   - First, we removed the exploration phase entirely (that is, the agent does not have a chance to obtain any information before it needs to exploit and achieve a goal). As expected, the agent performs very poorly without this phase (it learns a very simple strategy of intervening positively on the target node, which achieves consistent, but modest, rewards).
>   - Next, we replaced the exploration with a purely observational phase. Rather than allowing the agent to intervene to explore, it simply observes an equivalent number of samples from the DAG distribution without interventions. While the agent is able to use these observations to achieve slightly better performance, it still performs far worse than an agent that is allowed to intervene. This demonstrates the importance of exploration, and in particular the importance of allowing the agent to intervene at test time.
> * We have added some clarification of the explanations and how they are used in the RL environments (including examples in the text and a note in the figure caption), as well as in the LM experiments. We agree that the benefits of the explanations are not restricted to natural language explanations; rather it is the abstractions that they capture that are important, and we have noted this in the discussion.
> * We have added these references to the relevant portions of the discussion.

---

> > ### Comment · Reviewer_KP4B · 2023-08-11
> > **Thanks for you response**
> >
> > I have read the response and decide to keep my current score. I advocate for the acceptance of the paper

---

### Official Review · Reviewer_X6cj · 2023-07-06

**Soundness:** 3 good
**Presentation:** 3 good
**Contribution:** 4 excellent
**Rating:** 7
**Confidence:** 4

**Summary:**

This paper investigates whether it is possible for an agent to learn how to perform and make use of active causal interventions, despite being trained on purely passive / observational data (e.g. supervised learning, behavioral cloning, etc). The authors first argue that this is possible in principle, if the observed data contains behavioral traces of an expert who performs causal interventions in an environment to learn about its causal structure, and then exploits the learned information to achieve a goal. They then perform experiments in a causal DAG environment, a visual "odd-one-out" environment, and linguistic version of the "odd-one-out" environment (in order to test pretrained LLMs). In the casual DAG environment, they find that trained agents can learn to generalize causal exploration and exploitation strategies to unseen DAGs, just by passive learning of expert strategies on the training set. They find similar results in the odd-one-out environment, with stronger performance achieved when the training data includes natural language explanations of the interventions performed by the expert. Finally, they find that LLMs prompted with few-shot examples of causal experimentation can generalize to new settings where the latent cause of reward is absent from the few-shot examples, provided that explanation and reasoning traces are also provided in the few-shot examples.

**Strengths:**

This was a well-written paper that addresses an important topic of debate that has arisen about the capabilities of models trained purely on observational data, including pretrained (large) language models. I have several questions about the details of the experiments, and about the generality of causal experimentation strategies that are learned from observations of an expert, but overall I think the paper presents clear theoretical arguments and solid empirical demonstrations for its claims, showing that learning causal intervention strategies from offline/data is possible in principle, and is exhibited in practice to at least some degree of generality in actual ML models. The DAG experiments are especially well designed, showing that passively learned behavior is not well explained by approximate heuristics, and better explained by optimal causal reasoning. The odd-one-out and LLM experiments are more open to interpretation due to the absence of heuristic or correlational baselines, and not knowing whether the LLM was trained on similar data, but still provide evidence that passive learners may acquire generalizable causal intervention strategies in non-trivial settings (insofar as it is unlikely for the high performance to be achieved by a learned heuristic).

**Weaknesses:**

While the experiments have been designed carefully enough to show that observed casual strategies are generalized to some extent, I think they could be extended in the number of ways to probe the limits of this generalization.

In the DAG experiments for example, it is clear that the agent must have learned some kind of permutation invariant strategy for experimentation and intervention, given that the training data entirely lacks causal links that are present in the test episode. However, I think the experiments still leave open exactly how abstract the learned strategy is -- does it generalize to DAGs with 6 nodes, or exhibit size generalization more generally? (I expect not, due to the network architecture.) Can it generalize to DAGs with non-linear causal relationships? And can it adapt its experimentation strategy when it is *not* cued about which variables are relevant, perhaps by learning to imitate an expert that uses an optimal experimental design that minimizes the number of exploration queries before exploiting? (I found the explicit cues about which variables are relevant to be a fairly strong simplifying assumption -- it would be good to at least vary the number of variables which are cued as relevant to a wider range of numbers.) Similar size generalization tests would be interesting to test in the "odd-one-out" and LLM experiments -- e.g. having a test trial with 4 distinct objects, not just 3. For the LLM experiments in particular, it would also be good to say something about whether similar prompts are likely or unlikely to have appeared in Chinchilla 70B training data.

Generalizability is related to a more general question I have about what exactly "counts" as having learned an active causal strategy. In its strongest form, this might require verifying that a learned model actually implements a causal discovery algorithm and causal exploitation algorithm that is optimal, or at least sound and complete. Of course, this is very hard to verify without strong mechanistic interpretability techniques, and in the absence of that, heuristic baselines are helpful to compare against. Because heuristic baselines were included in the DAG experiments, I'm willing to believe that the model has successfully learned a close approximation to the optimal exploitation strategy in the n = 5 case, which presumably requires the agent to simulate the effect of a causal intervention on an internal representation of the DAG structure that was identified from the experimentation phase. But because similar baselines aren't included in the "odd-one-out" or LLM experiments, it could be that the learned strategy in those cases is just some "clever trick" that happens to do well most of the time without "true" causal reasoning. As such, it'd strengthen the paper to show some heuristic or correlational baselines for those experiments, if only to demonstrate that there are no obvious heuristics that can easily achieve the ~90% success rates in the odd-one-out and LLM experiments. For example, one baseline might be to always transform the color of the first/nearest object during experimentation (and to pick the unique colored object in the test trial if that transformation is successful). Another heuristic might be to always pick the object based on the feature that was most often unique in the training data. In addition, I think it'd good to note the limitations of what we can conclude about a model's learned strategies based on observational equivalence alone.

Regarding the need to predict/include linguistic explanations in the "odd-one-out" experiments, it was unclear to me whether the results shown in Figure 3 (and in the corrected version, Figure 13) are from a model that was also trained to predict natural language explanations. My guess is that they are, but I think the presentation clarity of this aspect could be improved, e.g. by stating in the caption of Figure 3 that the results shown involve predicting the explanation, by giving an example in the Appendix of what a full training episode (including the natural language explanations) looks like, and by explaining how the natural language explanations are generated (I had to consult the Lampinen et al (2022) paper in order to determine this).

Finally, I think it's somewhat misleading to describe the passive data in section 5.2.1 as "perfectly confounded" -- after all, the explanations constitute observational data that makes the setting no longer confounded! After all, it is only in worlds where some feature X causes reward R that choosing a object unique in feature X would result in the explanation E = e_X+ ("In this game, I was rewarded for unique X", and choosing an object unique in some other feature would result in the explanation E = e_X- ("The rewarding dimension must not be Y." where Y != X). Hence, the relevant part of the causal DAG is:

R <- X -> E

Once the agent observes E = e_X, it can reliably infer that there is causal link from X -> R in the true causal graph, since X is a common cause of both R and e_X. So the fact that explanations "deconfound" is no surprise at all.

**Questions:**

See above for various questions and suggestions I have about additional baselines, additional generalization tests, clarifying the role of explanations in the odd-one-out experiments, whether these models can be said to have "truly learned" causal reasoning, and whether the term "perfectly confounded" is apt.

In addition, I'm curious whether the authors still endorse the following analysis, given that their post-submission corrections show much better performance of passive learners in the odd-one-out experiment:

> Indeed, although our passive agents performed far above chance in the odd-one-out environments, they did not achieve as robust of performance as the online RL agents in earlier work [30]. This gap may be due to irreversible choices in these settings, which are difficult to learn from purely positive examples, without some active correction of off-optimal-policy behavior [e.g. 48, 11].

Post-Rebuttal Response:

The authors have addressed my questions and suggestions, and I continue to recommend acceptance.

**Limitations:**

Some limitations of their findings are discussed (see "What this paper does not imply:"), and the authors fairly summarize their findings as showing that passive learning of causal strategies is restricted to the case where expert intervention policies are actually present in the data. I think it'd be valuable to add discussion of exactly how much we can expect the learned causal strategies to generalize, and whether it is possible to establish if they actually constitute sound and complete causal discovery algorithms.

---

> ### Author Rebuttal · Authors · 2023-08-08
>
> We thank the reviewer for the positive assessment and for the thoughtful and extremely detailed comments and suggestions; we believe they will improve the paper. We have made a number of changes and performed some additional experiments in an attempt to address these comments:
>
> * While agents are architecturally limited to a fixed DAG size (which we have noted in an added limitations paragraph, see below) we performed two DAG generalization experiments, to further explore the robustness of the agents’ learned strategies. Results are plotted in the attached PDF.
>   - Generalizing to nonlinear edges: the nodes of the DAG already had nonlinearities in the main experiments, but at test time (only) we additionally added a nonlinearity on the edges (i.e. applied to each input before summing them in the node). Surprisingly, this had relatively little effect on evaluation performance (in retrospect, we believe this is because the post-node non-linearity masks the effect of the edge ones in many cases).
>   - Generalizing from training on linear DAGs to evaluation on non-linear DAGs. This is a harder generalization task, and agents correspondingly show a larger gap from optimal performance in evaluation. However, the agents still exhibit a fair amount of generalization (~75-80% of optimal performance).
>   - (We also performed a somewhat-relevant experiment exploring how reducing coverage of the training DAG distribution affects generalization to the original evaluation split, as requested by another reviewer.)
> * The design of the odd-one-out tasks, and the setup of our experiments, is such that simple heuristics should yield fairly low performance. We have added some discussion that tries to clarify the experiments, particularly with respect to the following points:
>   - Heuristics such as focusing on individual attributes, or always choosing the object that is unique by color, say, will not yield above-chance performance for the agents.
>   - For the LMs, the fact that the same prompts are yielding high performance along the “training” dimensions included in the prompt examples, and in evaluation on the other dimension, suggest that the model is not using a simple heuristic focusing on a single dimension (which would yield at most 55% performance).
>   - Furthermore, since we performed three separate splits — one where each feature dimension was held out — and observed similar performance across each, the results do not seem to be strongly influenced by the relative presence of the dimensions in the pretraining data (although without explanations, we do note that there are some differences among the dimensions in generalization).
> * In addition to the above, we added the suggested new experiment of evaluating the LLM with four test objects (not plotted in the PDF response due to space constraints). That is, we kept the prompts the same (three objects in the experimentation and test trials) but in addition to evaluating on a held-out dimension, we introduced a fourth object in the final test trial of the evaluation episode. More specifically, we kept three objects in the experimentation trials of the episode, and only surprised the model with a fourth object (which was not unique along any dimension) in the final choice of objects. While this does reduce generalization performance somewhat relative to the original three-object version, the performance is still far above chance (72% accuracy on average, chance is 25%; broken down by held-out dimension: Color:  66% +/- 8.5%, Shape: 71% +/- 12.6% Texture: 78.3% +/- 3.3%). This suggests that the LMs (when prompted with explanations) are relatively robust in generalization, even if the evaluation differs from the prompt in multiple ways.
> * We have made a number of additional edits to improve clarity and accuracy, per the reviewer’s suggestions (and some from the other reviewers), including:
>   - adding more detail about the explanations and their use.
>   - noting the limitations of the observational analyses rather than discovering computational strategy the agent is using.
>   - rephrasing the “perfectly-confounded” statements to clarify that the explanation removes the confounding.
>   - stating the fact that the agent could not generalize to DAGs with more nodes due to architectural limitations as part of a broader paragraph added to the discussion on limitations of the current environments, tasks, and agents.
>   - removing the statement about the gap with RL agents; indeed, our updated results no longer support the claim that it is a large difference. We have kept the more general point about DAGGER etc., however.

---

> > ### Comment · Reviewer_X6cj · 2023-08-12
> > **Thanks for the additional experiments.**
> >
> > This is great. It's nice to see the additional DAG and LLM experiments. I appreciate the explanation about how heuristics would perform in the odd-one-out experiments, and I think future readers will benefit from it as well. I continue to recommend acceptance.

---

### Official Review · Reviewer_c7Wr · 2023-07-07

**Soundness:** 3 good
**Presentation:** 3 good
**Contribution:** 3 good
**Rating:** 7
**Confidence:** 4

**Summary:**

The paper deals with the problem of understanding whether causal structures can be learned from passive observational data. The study shows formally that it is possible for an agent to learn a generalizable strategy from passive data, as long as the agent can intervene at test time. Then, the experimental results show empirically that agents trained on expert data can indeed generalize at test time to infer and use causal links which are never present in the training data.

**Strengths:**

* The paper addresses an interesting and important problem
* The paper shows that natural language explanations support generalizing causal strategies from passive data
* The findings of the paper have several applications, including using tools with LLMs
* The paper is well written and easy to follow

**Weaknesses:**

* It would be interesting to explore more complex environments

**Questions:**

None

**Limitations:**

The paper should describe limitations more explicitly in a dedicated paragraph

---

> ### Author Rebuttal · Authors · 2023-08-08
>
> We greatly appreciate the positive comments. We have added a paragraph to the discussion noting some of the limitations of the environments and tasks used, since these limitations were noted by several reviewers. Although we have not been able to experiment with substantially more complex environments during the brief response period, we have performed a few experiments that slightly increase the challenge of the present tasks (such as generalizing to different types of causal graphs at test time, or reducing coverage of the training distribution), as requested by other reviewers. The results can be found in the global response PDF.

---

### Author Rebuttal · Authors · 2023-08-08

We appreciate the thoughtful and thorough reviews. We are pleased that the reviewers found the paper “interesting” and “important” and felt that it addresses a “fundamental question,” and is “well-written” and “sound.” We have tried to clarify and refine the points noted by the reviewers (see individual responses for more details). In the process, we have run several of the suggested ablations and generalization experiments to clarify the results:
* Ablations that explicitly demonstrate the importance of allowing an interventional exploration phase at evaluation time. If we remove the active (interventional) exploration phase, either replacing it with a purely observational phase (where the agent simply sees samples from the DAG distribution), or removing it entirely, the agent performs much worse. (Fig. 13 in the response PDF.)
* Exploring how causal strategy generalization depends on the variety of causal DAG structures in training, by running two additional experiments that hold out successively more DAG structures. If we reduce the number of possible DAG structures to about 33% of the original training set, the agents still generalize fairly well, but if we reduce it to only 15%, the agents perform much worse. (Fig. 14 in the response PDF.)
* Evaluating the ability of agents to generalize their causal strategies to DAGs with different functional structures at evaluation time, either by testing with additional nonlinearities on the edges, or by training the agents on linear DAGs and testing on nonlinear ones. In either case we see reasonable generalization, though the linear -> nonlinear case is much harder. (Fig. 15 in the response PDF.)
* Evaluating the LLM with a surprise extra test object. That is, we kept the prompts the same (three objects in the experimentation and test trials) but in addition to evaluating on a held-out dimension, we introduced a fourth object in the final test trial of the evaluation episode. More specifically, we kept three objects in the experimentation trials of the episode, and only surprised the model with a fourth object (which was not unique along any dimension) in the final choice of objects. While this does reduce generalization performance somewhat relative to the original three-object version, performance is still far above chance (72% accuracy on average, chance is 25%; broken down by held-out dimension: Color:  66% +/- 8.5%, Shape: 71% +/- 12.6% Texture: 78.3% +/- 3.3%). This suggests that the LMs (when prompted with explanations) are relatively robust in generalization, even if the evaluation differs from the prompt in multiple ways.  (Not plotted in the PDF due to space constraints.)

We believe these experiments, and the edits suggested by the reviewers, have improved the paper, and we thank the reviewers for their valuable suggestions.

---

### Decision · Program_Chairs · 2023-09-21

**Decision:**

Accept (poster)

**Comment:**

This paper proposes an interesting strategy to improve generalization. The ideal is to motivate agents to learn a policy that actively learns the causal structure in the data on deployment, followed by reward optimizing behavior and this policy results in better generalization by virtue of having learned causal dependencies between variables not observed in training.

The overall reception to this paper has been positive. Authors also conducted additional ablations, which I appreciated on the generalization quality vs number of causal DAGs used in training. The interesting aspect is the it does not deteriorate as fast as I expected.

My own criticism of the paper is mainly that I believe it lacks precision in the claims it is making. While empirical performance is great, the paper doesn't answer exactly how the causal structure is benefiting the exploitation phase. I would also have liked more associational baselines.

Considering the overall positive response, I recommend an accept and encourage the authors to include a more nuanced discussion in the camera ready version.